

# A methodology for estimating the response of the coastal ocean to meteorological forcing: A case study in the Bohai Bay

Daosheng Wang[1,2,3], Haidong Pan[3], Lin Mu[1,2], Xianqing Lv[3], Bing Yan[4], Hua Yang[4]

[1]College of Marine Science and Technology, China University of Geosciences, Wuhan 430074, China
[2]Shenzhen Research Institute, China University of Geosciences, Shenzhen 518057, China
[3]Physical Oceanography Laboratory/CIMST, Ocean University of China and Qingdao National Laboratory for Marine Science and Technology, Qingdao 266100, China
[4]Key Laboratory of Engineering Sediment of the Ministry of Transport/National Engineering Laboratory for Port Hydraulic Construction Technology, Tianjin Research Institute for Water Transport Engineering, M.O.T., Tianjin 300456, China

*Correspondence to*: Lin Mu (moulin1977@hotmail.com)

**Abstract.** The sea level (SL) variations at the coastal ocean result from multiscale processes and are substantially contributed by the SL changes due to the meteorological forcing. In this study, a new methodology, named as IBR, is developed to estimate the response of the coastal ocean to meteorological forcing. The response is taken as the combination of the static ocean response calculated using the inverted barometer formula and the dynamic ocean response estimated using

the multivariable linear regression involving atmospheric pressure and wind component at the dominant wind orientation. The dominant wind orientation is determined based on the averaged values of the magnitude squared coherences between the adjusted SL and wind at every wind orientation.

The IBR is implemented to estimate the response of the coastal ocean at two stations, E1 and E2 in the Bohai Bay, China. The analysed results indicate that at both E1 and E2, the adjusted SLs are related more to the regional wind, which is

the averaged value in the Bohai Bay of the 10 m wind in the ERA-Interim data, than to the local wind; the dominant regional wind orientation is 75°. The estimated response using IBR with the regional meteorological forcing is much closer to the observed values than other methods, including the classical inverted barometer correction, the dynamic atmospheric correction, the multivariable linear regression and the IBR with local forcing, demonstrating that IBR with regional forcing have the best skill in estimating the response. The large deviations between the observed values and the estimated values

using IBR with the regional meteorological forcing are mainly due to the remote wind, which is not considered in the IBR. This case study indicates that the IBR is a feasible and relatively effective method to estimate the response of the coastal ocean to the meteorological forcing.

## 1 Introduction

The elevated sea level (SL, see Appendix A) at the coast is dangerous for the nearby city, while the depressed sea level

can render navigation of coastal bays and harbors difficult and hazardous (Tilburg and Garvine, 2004). The SL variations at the coast result from multiscale processes, with the superposition of global mean SL, regional SL and local SL changes, as





shown in Melet et al. (2016) and Melet et al. (2018). SL changes due to the meteorological forcing, including sea surface atmospheric pressure (AP) and wind, make substantial contributions to total SL changes in coastal ocean (Melet et al., 2018; Zhang et al., 2019). Traditionally, the ocean response to meteorological forcing is estimated simply using the inverted barometer (IB) correction, which makes the response be poorly accounted for (Wunsch and Stammer, 1997). The classical

IB formulates the static response of the ocean to AP forcing, and the wind effects are totally ignored (Carrère and Lyard, 2003). However, the coastal ocean response is different from the IB response in shallow water and the influence of the wind forcing cannot be ignored (Carrère and Lyard, 2003; Lv et al., 2018).

The dynamical atmospheric correction (DAC), which is a combination of the high frequencies of simulated response forced by AP and wind using numerical model and the low frequencies of the IB correction (Carrere et al., 2016), has been

widely implemented to estimate the barotropic response of the global ocean to meteorological forcing, especially in the altimetry SL estimations. Tierney et al. (2000) pointed out that the simulated SL signals at periods less than 20 days, which were not uniform in space but can be enhanced in some semi-enclosed regions (Fukumori et al., 1998), were more consistent with observations when using AP plus wind forcing than using wind forcing alone. Hirose et al. (2001) used a barotropic, shallow water model to simulate the response of the ocean to atmospheric disturbances and found that the AP-driven results

accounted for a small portion of the observed SL variance while the wind-driven results explained a large amount of the variability in the observations at middle and high latitudes. Carrère and Lyard (2003) simulated the global ocean response to atmospheric wind and pressure forcing using the MOG2D-G model, and found the model correction could reduce the SL variance at high latitudes, continental shelf areas and shallow waters compared to the classical IB correction. Melet et al. (2016) evaluated the relative importance of processes causing coastal SL variability at different time-scales, in which the SL

induced by AP and wind was estimated using the DAC.

The regression analysis was also widely implemented to investigate the response of oceans to the meteorological forcing. Tilburg and Garvine (2004) developed a simple linear-regression model based on modest wind forecast capability and records of local coastal sea level, wind and pressure, and found that this empirical model was adequate for general use. Based on the results of regression analysis, Andres et al. (2013) hypothesized that the annual SL changes along the shoreline

were significantly influenced by the local winds and barotropic response. Calafat and Chambers (2013) found that a multivariable linear regression (MLR) involving local wind and AP can account for a substantial fraction of the annual SL variation at the Boston and New York tide gauges. When the MLR involving the local wind and AP was implemented, Lv et al. (2018) found that the response of the coastal ocean to AP had the similar form to the IB correction and the estimated results were much closer to the observations than those using only wind or IB correction or combination of them.

In the DAC, the high frequencies of simulated response forced by AP and wind and the low frequencies of the IB correction represent the dynamic ocean response and the static ocean response, respectively (Piecuch et al., 2016). It is noted that the fine and accurate topography data in the coastal and shallow area is difficult to be obtained, which strongly influences the wind-driven barotropic SL variability (Fukumori et al., 1998), so the dynamic ocean response simulated by numerical model may be not accurate in the coastal and shallow area. Moreover, the static ocean response is always ignored





in the regression analysis. Besides, the SL change may be not determined by the local meteorological forcing, as shown in Amin (1982), Thompson et al. (2014) and Piecuch et al. (2016).

The purpose of this paper is to present a new methodology, in which the response of the coastal ocean to the meteorological forcing is estimated using the combination of the static ocean response, which is calculated using IB correction, and the dynamic ocean response, which is estimated using regression analysis, and to demonstrate the feasibility and effectivity of this new method. Besides the local meteorological forcing as shown in Lv et al. (2018), the influence of the regional meteorological forcing will also be considered in this study; in addition, the low-pass wind and AP are used in the regression analysis in this study, rather than the original forcing as used in Lv et al. (2018). The details of the rest of the paper are as follows: the new methodology and data are shown in section 2; the estimated response of the coastal ocean to the meteorological forcing in the Bohai Bay is shown in section 3; the comparisons with the other methods and the reason for the large deviations in the estimated results are discussed in section 4; and the conclusions can be found in section 5.

## 2 Methodology and Data

### 2.1 Methodology for estimating the response of the coastal ocean to meteorological forcing

As shown in Sandstrom (1980), Hsueh and Romea (1983), Castro and Lee (1995) and Lv et al. (2018), the response of the coastal ocean to meteorological forcing is mainly at low frequencies and is the combination of the following two main components (Piecuch et al., 2016):

$$h_L = h_{static} + h_{dynamic} \tag{1}$$

where, $h_L$ is the low-frequency SL, and $h_{static}$ and $h_{dynamic}$ indicate the static ocean response and the dynamic response, respectively.

According to the DAC, the static ocean response can be estimated using IB correction. However, the dynamic ocean response in the DAC, estimated using the simulated results by numerical model, may not be accurate in the coastal and shallow area, as the resolution of topography data is not always high enough; on the contrary, the regression analysis is not affected by the resolution of topography data. The response of the ocean to meteorological forcing, including AP and wind, is approximately linear (Li and Yang, 1983), as demonstrated in Calafat and Chambers (2013) and Lv et al. (2018), so the dynamic ocean response can be estimated using the linear combination of AP and wind. As the static and dynamic ocean responses are linear in the meteorological forcing, the low-frequency AP and low-frequency wind will be used to estimate the ocean response to avoid introducing the high-frequency signals into the estimated low-frequency SL. Therefore, the response of the coastal ocean to meteorological forcing is estimated as follows:

$$h_L = h_{static} + h_{dynamic} = h_{IB} + \alpha_0 + \alpha_1 \times \mathrm{LowAP} + \alpha_2 \times \mathrm{LowWind} \tag{2}$$





where, LowAP is the low-frequency AP; LowWind is the low-frequency wind component at the dominant wind orientation, which can be determined based on the result of magnitude squared coherence, as shown in Lv et al. (2018); $h_{IB}$ is the SL estimated using the IB formula with LowAP; $\alpha_0$ is a constant, $\alpha_1$ and $\alpha_2$ are coefficients.

As the IB formula is explicit, as described in Paraso and Valle-Levinson (1996) and Kurapov et al. (2017), the linear combination of low-frequency AP and low-frequency wind is the adjusted SL (ASL) in fact and is equal to the difference between low-frequency SL and IB correction. $\alpha_0$, $\alpha_1$ and $\alpha_2$, can be solved in a least squares sense by regressing the ASL onto the low-frequency wind component at the dominant wind orientation and low-frequency AP:

$$h_{dynamic} = h_L - h_{IB} = \alpha_0 + \alpha_1 \times \text{LowAP} + \alpha_2 \times \text{LowWind} \qquad (3)$$

When $\alpha_0$, $\alpha_1$ and $\alpha_2$ are determined, the response of the coastal ocean to the meteorological forcing can be estimated using Eq. (2). As both the classical IB correction and regression analysis are implemented, this new methodology is labelled as IBR hereafter.

## 2.2 In-situ observations

In-situ observations were obtained from two stations, E1 and E2, in the Bohai Bay (Fig. 1). At each station, the total SL was measured using the moored pressure gauge, which was accurate to within 5 cm (Lv et al., 2018); in addition, the meteorological observations, including 10 m wind and sea level AP, were measured using the XZY3-type automatic observing system produced by the National Ocean Technology Center, SOA of China, of which the 10 m wind was measured by the XFY3-1 propeller anemograph that had been widely used in most coastal station systems in China (Wang et al., 2015). The XFY3-1 propeller anemograph was accurate to within 5° for wind direction and 0.5 m/s for wind speed (Lv et al., 2018). The hourly in-situ observations at E1 and E2 started at 0000 UTC 19 August and ended at 0000 UTC 18 November 2014. These meteorological observations were taken as the local forcing in this study.

Based on the IB correction (Paraso and Valle-Levinson, 1996), adjustment was made to the hourly total SL using the observed AP, and ASL was obtained. Similar to Lv et al. (2018), the ASLs were filtered using a cosine-lanczos filter (Duchon, 1979) with a high frequency cut-off of 0.8 cpd, through which the tidal, local inertial and higher frequency signals were eliminated, and the low-pass result was labelled as LASL. The wind components were similarly filtered, and the low-pass $u$ wind, $v$ wind and wind speed were labelled as LUW, LVW and LW, respectively. The similarly filtered total SL and AP were labelled as LSL and LAP, respectively. Similar to Sandstrom (1980), Hsueh and Romea (1983), Castro and Lee (1995) and Lv et al. (2018), all the hourly low-pass data, LASL, LUW, LVW, LW, LSL and LAP, were sub-sampled at 6-hourly intervals, and the results were labelled as SLASL, SLUW, SLVW, SLW, SLSL and SLAP, respectively. The time series of SLUW, SLVW, SLSL and SLAP at E1 and E2 are shown in Fig. 2.



### 2.3 ERA-Interim data

ERA-Interim data is the global atmospheric reanalysis data produced by the European Centre for Medium-Range Weather Forecasts (Dee et al., 2011). The ERA-Interim data is from 1 January 1979 to present and the minimum time interval is 6 hours. The details about the ERA-Interim data can be found in Dee et al. (2011). Six-hourly ERA-Interim analysis data of AP and 10 m wind, with a horizontal resolution of 0.125° latitude and longitude, were used to calculate the regional meteorological forcing in this study.

To evaluate the quality of the ERA-Interim data, the six-hourly AP and wind components in the ERA-Interim data and the sub-sampled in-situ observations, at E1 and E2, are compared in Fig. 3. It can be seen that the magnitudes and trends were similar to each other. The mean absolute errors (MAEs) and the correlation coefficients between the meteorological forcing in the sub-sampled in-situ observations and those in the ERA-Interim data are listed in Table 1. The MAEs for AP at both E1 and E2 were less than 0.85 mbar, indicating that AP in ERA-Interim data was close to the observed values in the in-situ observations, as shown in Fig. 3a and Fig. 3d. The MAEs for $u$ wind and $v$ wind at E1 were about 1.6 m/s, but the correlation coefficients were larger than 0.8, indicating that both the $u$ wind and $v$ wind in the ERA-Interim data at E1 were in good agreement with the observed values in the in-situ observations, as shown in Fig. 3b and Fig. 3c. The same conclusions can be obtained at E2. Overall, the meteorological forcing data in ERA-interim data agreed well with the in-situ observations and can be used to calculate the regional forcing.

The regional meteorological forcing in the Bohai Bay, including AP and 10 m wind, was defined as the averaged value over the region spanning 117.5°-119°E and 37.9°-39.3°N as shown by the area R1 in Fig. 1b. The regional meteorology was then similarly filtered with a high frequency cut-off of 0.8 cpd. The low-pass results were labelled as ERA-LAP and ERA-LW, respectively. The difference between the sub-sampled in-situ observed SL and the IB correction calculated using ERA-LAP was taken as the ERA-ASL, which was also filtered with a high frequency cut-off of 0.8 cpd, and the result was labelled as ERA-LASL.

### 3 Results

### 3.1 Relationship between adjusted sea level and local wind

Following Lv et al. (2018), the magnitude squared coherences between SLASL and SLW were calculated, as functions of frequencies and wind orientation measured clockwise from north at an interval of 5 degrees, to find the relationship between the local ASL (i.e., SLASL) and local wind (i.e., SLW). The calculated results at E1 and E2 are shown in Fig. 4a and Fig. 4d, respectively. The corresponding averaged values of the magnitude squared coherences are shown in Fig. 4b and Fig. 4e for E1 and E2, respectively. It can be seen that the averaged magnitude squared coherences reached the maximum when the wind orientation was 80° at E1 and 10° at E2, demonstrating that the dominant wind orientations were 80° and 10° at E1 and E2, respectively, which were different from those at which the maximum magnitude squared coherences were



obtained (Fig. 4a and Fig. 4d). The wavelet coherence and phase (Grinsted et al., 2004) between SLASL and SLW at the dominant wind orientation at E1 are shown in Fig. 4c, and those at E2 are shown in Fig. 4f. It was indicated that at E1, the SLASL and the SLW at 80° were antiphase in most of the periods at the 5% significance level, especially at 8-16 days. At E2, the lag time between SLW at 10° and SLASL was much larger than that at E1 (Fig. 4f).

As shown in Fig. 2, the SLSLs at E1 and E2 were almost equal, but the SLUW and SLVW at E1 were far from those at E2. Besides, the correlation coefficient between SLASL at E1 and the SLW at the dominant wind orientation (80°) was -0.56, while it was just -0.17 at E2. It was guessed that the SLASL at E2 may be more related to the SLW at E1 than that at E2. The magnitude squared coherences between SLASL at E2 and SLW at E1 were calculated and are shown in Fig. 4g; besides, the averaged values at each wind orientation are shown in Fig. 4h. The patterns of Fig. 4g and Fig. 4h are similar to Fig. 4a and

Fig. 4b, respectively, indicating that the E2 SLASL response to E1 SLW was similar to the E1 SLASL response to E1 SLW. The maximum value of the averaged magnitude squared coherences between SLASL at E2 and SLW at E1 was 0.41, which was much larger than that between SLASL at E2 and SLW at E2; moreover, the dominant wind orientation was 85°, which is similar to that shown in Fig. 4b; besides, the relationship between SLASL at E2 and SLW at E1 was negative and the lag time was much smaller than that between SLASL at E2 and SLW at E2 (Fig. 4i). It was hypothesized from the

aforementioned results that the wind influencing the SLASL at E2 may be not the local wind, but the regional wind.

### 3.2 Relationship between adjusted sea level and regional wind

   The magnitude squared coherences between the regional ASL (i.e., ERA-LASL) and regional wind (i.e., ERA-LW), at E1 and E2, were calculated and are shown in Fig. 5a and Fig. 5d, respectively. At both E1 and E2, the periods at which the coherences reached the maximum value were much less than those between SLASL and SLW, while the variations of the

averaged values (Fig. 5b for E1 and Fig. 5e for E2) were similar to those shown in Fig. 4. Based on the averaged value of the magnitude squared coherences, the dominant wind orientation influencing ERA-LASL, at both E1 and E2, was 75°, further indicating that the regional wind affected the ASL in the Bohai Bay. Besides, at both E1 and E2, the ERA-LASL and ERA-LW at 75° were nearly antiphase in most of the periods at the 5% significance level (Fig. 5c and Fig. 5f), showing that the relationship between them was negative.

The correlation coefficient between the ERA-LASL at E1 and ERA-LW at 75° was -0.62, which was much larger than that between SLASL at E1 and SLW at 80° (-0.56). The correlation coefficient between SLASL at E2 and SLW at 10° was just -0.17 and that between SLASL at E2 and SLW at 85° at E1 was -0.54, which were less than that between ERA-LASL at E2 and ERA-LW at 75° (-0.58). The aforementioned results indicated that the ASL, at both E1 and E2, had much stronger relationship with the regional wind than local wind, providing guidance for estimating the response of the coastal ocean to

the meteorological forcing in the Bohai Bay.





### 3.3 Response of the coastal ocean to meteorological forcing in the Bohai Bay

Based on the aforementioned conclusions, the regional meteorological forcing was the dominant driver of low-frequency SL in the Bohai Bay, and will be used in the IBR to estimate the response of the coastal ocean to meteorological forcing. The ERA-LASLs at E1 and E2 were regressed onto the ERA-LW at 75° and ERA-LAP, as follows:

$$\text{ERA-LASL} = \alpha_0 + \alpha_1 \times \text{ERA-LAP} + \alpha_2 \times \text{ERA-LW} \tag{4}$$

The regression coefficients in Eq. (4) are listed in Table 2. It was noted that the MLR with Eq. (4) at both E1 and E2 were significant ($p<0.001$). It was apparent from the regression coefficients in Table 2 that ERA-LASLs at E1 and E2 were negatively correlated with both ERA-LAP and ERA-LW, as both $\alpha_1$ and $\alpha_2$ were negative; besides, the dynamic responses of the ocean at E1 to both ERA-LAP and ERA-LW at 75° were larger than those at E2, as $\alpha_1$ and $\alpha_2$ at E1 were larger than those at E2, which may be because the water depth at E1 was much smaller than that at E2, as shown in Fig. 1b. When ERA-LAP increases by 1 mbar, the SLSL will decrease by $1.94\times10^{-2}$ m at E1 and $1.78\times10^{-2}$ m at E2, with both the static ocean response and the dynamic ocean response considered, which were of the same order as those in Lv et al. (2018), suggesting that MLR in IBR was reasonable.

Combined the IB correction using ERA-LAP and the result of MLR with Eq. (4), the SLSL at E1 was estimated using Eq. (2) and is shown in Fig. 6a, while the result at E2 is shown in Fig. 6b. At both E1 and E2, the estimated SLSLs reproduced most of the local maxima of the observed values, although the observations were slightly underestimated in some cases; however, the estimated SLSLs deviated greatly from the observed values at most of the local minima. In addition, the SLSLs at both E1 and E2 were mainly attributed to the SL variabilities resulting from regional wind (i.e., ERA-LW), rather than AP (i.e., ERA-LAP), as shown in Fig. 6. The correlation coefficient and MAE between the observed and estimated SLSL at E1 (E2) were 0.70 (0.65) and 13.13 cm (13.44 cm), respectively, as listed in Table 3. Following MWRPRC (2014), Kurapov et al. (2017) and Lv et al. (2018), the frequency of occurrences when the estimated SLSL was within 0.15 m from the observed SLSL, labelled as FO, was taken as another metrics for the evaluation. FOs at E1 and E2 were 72.33% and 71.78%, respectively, as listed in Table 3. The results indicated that the estimated responses of the coastal ocean to the meteorological forcing, using IBR with regional forcing, could account for a significant fraction of the observed SLSLs.

## 4 Discussions

### 4.1 Comparison with the classical IB correction

Traditionally, the IB correction was the classical method to estimate the response of the coastal ocean to the meteorological forcing (Wunsch and Stammer, 1997). As listed in Table 3, when the IB correction with local SLAP was used to estimate the response at E1 (E2), the MAE between the estimated SLSL and the observed values was 20.13 cm (19.28 cm), which was much larger than that when IBR with regional forcing was implemented; besides, FO was only 48.22% (47.40%), showing that the estimated SLSLs was far from the observed values on the whole, as shown in Fig. 7a (Fig. 8a).





Comparison of the SLSLs at E1 (E2) estimated using the IB correction and IBR with regional forcing showed that IBR with regional forcing can improve FO by as much as additional 24.11% (24.38%). The aforementioned results indicated that the classical IB correction cannot accurately estimate the SLSL and the dynamic ocean response cannot be ignored.

## 4.2 Comparison with the dynamic atmospheric correction

As the ocean has dynamic response to meteorological forcing at high frequencies when considering large scales (Vinogradova et al., 2007), the DAC can significantly improve the altimetry product compared with the classical IB correction. To compare the estimated results using the IBR with regional forcing and the DAC in the Bohai Bay, MITgcm (Marshall et al., 1997) was used to simulate the dynamic response of the ocean to the meteorological forcing at E1 and E2. The computing area was the whole Bohai Sea, as shown in Fig. 1. The six-hourly AP and 10 m wind, extracted from the
ERA-Interim data, were interpolated spatially and temporally to obtain the surface forcing. Flather radiation condition was used at the east open boundary, which is shown in Fig. 1b. As the meteorological forcing, model dimension, horizontal resolution and vertical stratification may affect the simulated results, several experiments (Exp1-Exp10) were carried out to discuss the factors. The detailed model settings of the numerical experiments are listed in Table 4. In all the above experiments, the model began with a cold start from 0000 UTC 1 June 2014, and continued running with a time step of 60s
until 0000 UTC 20 November 2014. The initial 78 days were used for spin-up.

       For validation of the numerical model, ocean tides, which were similar to atmospheric forced ocean waves (Carrère and Lyard, 2003), were simulated using MITgcm with the similar model settings. Four dominating constituents, including $M_2$, $S_2$, $K_1$ and $O_1$, were implemented as tidal forcing at the east open boundary (Fig. 1b); the tidal information was extracted from Oregon State University Tidal Inversion Software (Egbert and Erofeeva, 2002). For the sake of simplicity, the model settings
were similar to those in Exp1-Exp4, in which the model was two dimensional and the constant TS1 profile in Fig. 9 was used. The horizontal resolution was 7.5′×7.5′ in Exp1-tide and 2′×2′ in Exp2-tide, respectively. The harmonic constants of each constituent, analysed from the simulated water level of last 30 days from 0000 UTC 1 August 2014, were compared with the observations at 14 tide gauge stations, whose locations are shown in Fig. 1b. Whether in Exp1-tide or Exp2-tide, the MAEs between the simulated and observed amplitudes of all the constituents were less than 10 cm, as shown in Fig. 10, while the
MAEs for the phase lag were less than 16°, showing that the numerical model was validated and acceptable.

       Following Carrère and Lyard (2003) ,the residual signal variance and the ratio of the variation reduction compared with the IB correction, at E1 and E2, were calculated in all the numerical experiments and are listed in Table 5. When only AP was taken as the meteorological forcing, the residual signal variations at E1 (E2) were reduced less than 1.31% (1.84%), whatever the dimension, horizontal resolution and vertical stratification were assigned, showing that the DAC with only AP
led to better estimated result than that using the IB correction, but the improvement was slight, similar to the conclusions in Hirose et al. (2001). On the contrary, the AP plus wind forcing reduced the variance at E1 (E2) by 23.12% (19.79%) to 24.57% (20.71%), further showing that the wind was an important driver of SLSL and cannot be ignored.





Hirose et al. (2001) concluded that the fine topography was preferable, but they only used a two-dimensional barotropic ocean model. The same conclusion can be drawn from the comparison of the simulated results in Exp2 and Exp4. However, when the simulated results in Exp8 and Exp10 were compared, it can be seen that the simulated response of sea level to AP and wind at E2 was improved when the topography was finer, but the results at E1 were not, which may be because the

horizontal resolution of the ERA-Interim meteorological forcing was just 7.5′.

Tierney et al. (2000) concluded that the density stratification did not make much difference in modelling SL high frequency signals (periods shorter than several weeks) because the response was essentially barotropic, which was also shown in Vinogradova et al. (2007). The results in Exp5 and Exp6 indicated that the near real density stratification TS2 did not improve the performance of DAC, when AP was taken as the meteorological forcing. Improvement was not observed

either, when comparing Exp7 and Exp8 where both AP and wind were considered. However, when the density stratification TS1 was used, the residual variance can be further decreased if the model was changed to three dimensional from two dimensional in vertical direction, as can be seen when the results in Exp2 and Exp7 (Exp1 and Exp5, or Exp3 and Exp9) were compared.

Considering the sum of the ratio of the variation reduction at E1 and E2, the best performance of DAC was obtained in

Exp7, in which both AP and wind were included in the forcing and the variation reduction reached 24.57% at E1 and 20.30% at E2 when compared with the IB correction, while the estimated results using the IBR with regional forcing induced a greater reduction of 42.27% at E1 and 36.79% at E2; besides, the IBR with regional forcing can improve FO by as much as additional 20.82% at E1 and 17.53% at E2, when compared to the results of DAC in Exp7, as listed in Table 3. The results indicated that the IBR with regional forcing had much better performance than the DAC, as shown in Fig. 7b and Fig. 8b.

**4.3 Comparison with the multivariable linear regression**

Based on the same observations used in this study, Lv et al. (2018) compared several regression methods and found that the best estimated results were obtained when the MLR involving wind component at dominant orientation and AP was used, as follows:

$$SLSL = \gamma_0 + \gamma_1 \times SAP + \gamma_2 \times SW \tag{5}$$

where SAP and SW are the sub-sampled AP and wind component at the dominant wind orientation, respectively; $\gamma_0$ is a constant, $\gamma_1$ and $\gamma_2$ are coefficients.

As listed in Table 3, the MAEs between the estimated SLSL in Lv et al. (2018) and the observed values were 14.51 cm at E1 and 16.03 cm at E2, which were much less than those when IB correction or DAC were used; besides, FOs at E1 and E2 were also larger than those with IB correction or DAC, showing that MLR with Eq. (5) was a useful method to estimate

the response.

However, it was obvious that the estimated results using MLR with Eq. (5) at E1 and E2 were much farther from the observations than those using IBR with regional forcing, as shown in Table 3, Table 5, Fig. 7a and Fig. 8a. As there were





some high frequency signals in the original wind, which will introduce significant harmonic motion into water bodies (Militello and Kraus, 2001), the estimated SLSLs using MLR with Eq. (5) included much more high frequency signals than those using IBR with low-pass regional forcing, at E1 (Fig. 7a) and E2 (Fig. 8a). Overall, the IBR with regional forcing had much better performance than MLR with Eq. (5) presented in Lv et al. (2018).

**4.4 Comparison with the IBR using local meteorological forcing**

For IBR with local meteorological forcing implemented, the MLR is as follows:

$$\text{LASL} = \alpha_0 + \alpha_1 \times \text{LAP} + \alpha_2 \times \text{LW} \tag{6}$$

The regression coefficients are listed in Table 2. When the regression coefficients were determined, the estimated SLSLs at E1 (E2) can be obtained using Eq. (2) and are shown in Fig. 7b (Fig. 8b). The estimated SLSL at E2 was slightly farther from the observed values than that using MLR with Eq. (5) (Table 3), as the correlation coefficient between SLASL and local wind at E2 was too small; however, the estimated SLSL at E1 was slightly closer to the observed values than that using MLR with Eq. (5), showing that the IBR with local forcing was not always worse than MLR with Eq. (5). As shown in Table 3, at both E1 and E2, the estimated SLSLs using IBR with regional meteorological forcing were much closer to the observed values than those using all the other methods, including IBR with the local forcing, demonstrating that IBR with regional forcing had the best skill in estimating the response of the coastal ocean to the meteorological forcing in the Bohai Bay.

**4.5 Discussion about the response using IBR with regional meteorological forcing in the Bohai Bay**

As the dominant wind orientation was 75° and the relationship between ERA-LASL and ERA-LW at 75° was negative, at both E1 and E2, the regional across-shore wind was an important factor influencing the ASL, which may be related with the fact that the Bohai Bay is a part of the Bohai Sea, a semi-enclosed coastal sea; in detail, the onshore wind will increase the ASL and the offshore wind will decrease the ASL, the same as in Lv et al. (2018) and different from continental shelves (Andres et al., 2013; Hsueh and Romea, 1983; Zhao and Cao, 1987). Besides, as shown in this study, the regional meteorological forcing was the dominant driver of SLSL in the Bohai Bay, rather than the local forcing, indicating that the dominant factors should be determined carefully at a specific location.

Although the estimated results using IBR with regional forcing were much closer to the observed response than those using other methods, including IB, DAC, MLR with Eq. (5) and IBR with local forcing, at both E1 (Fig. 7) and E2 (Fig. 8), it should be noted that the estimated results were far from the observations during some extreme events, as shown in Fig. 6. Spitz and Klinck (1998) pointed out that the response of the sea water to wind forcing in the Bay was complex, depending on local forcing and nonlocal forcing. As shown in Fig. 11, before the SLSL reached the local minimum at yearday 306 (316), the wind was much stronger than 10 m/s and in NW-SE direction in the north Yellow Sea and Bohai Strait from yearday 305.25 (315.25) to 305.5 (315.5), reducing the SLSL locally due to wind set-up and may make the SLSLs at E1 and E2





extreme at yearday 306 (316) due to swell swash (Melet et al., 2018). As shown in Fig. 12, before the SLSL reached the local maximum at yeardays 277.25 and 284, the wind was not strong; however, the wind in the Bohai Sea at yearday 284 was much stronger than 10 m/s, while the ERA-LW at 75° was just about -5 m/s, indicating that the wind set-up in the Bohai Sea may be the cause of this extreme event at E1 and E2 and a smaller lag time between SLSL and wind than that when the

SLSL reached it local minima at yeardays 306 and 316. On the contrary, the wind during yearday 276.5 to 277.25 was not stronger in the Bohai Sea and north Yellow Sea, as shown in Fig. 12, so extreme SLSL at yearday 277.25 was not related with the wind and the cause was not clear. Overall, except the extreme maximum at yearday 277.25, the other three extreme events at yeardays 284, 306 and 316 were influenced by the remote wind that was not considered in the IBR, and therefore the estimated results using IBR with regional forcing during these extreme events deviated greatly from the observed SLSLs

at both E1 and E2. The aforementioned results indicated that besides the regional meteorological forcing, there were other known and unknown factors influencing the SLSL in the Bohai Bay, which should be further investigated in the future.

## 5 Conclusions

The response of the coastal ocean to the meteorological forcing was a significant part of the total SL variations and cannot be estimated accurately using the traditional IB correction, as shown in Carrère and Lyard (2003) and Lv et al. (2018).

Building on the description of the static ocean response and dynamic ocean response in Piecuch et al. (2016), the DAC in Carrere et al. (2016) and the MLR with Eq. (5) in Lv et al. (2018), a new methodology, named as IBR, was developed in this study to estimate the response of the coastal ocean to the meteorological forcing. In IBR, the response was a combination of the static ocean response and the dynamic ocean response. The former component was calculated using IB formula and the latter component was estimated using MLR with Eq. (3) involving low-pass AP and wind component at the dominant wind

orientation. The observed SL from two stations, E1 and E2 located in the Bohai Bay, China, were taken as a case study to evaluate the feasibility and effectivity of IBR and compare it with other methods.

The magnitude squared coherences between SLASL and SLW were calculated and the averaged values at every orientation were taken as the indicator to find the dominant wind orientation. It was found that the correlation coefficient between SLASL at E2 and SLW at the dominant wind orientation at E1 (R=-0.54) was much larger than that between

SLASL at E2 and SLW at the dominant wind orientation at E2 (R=-0.17), indicating that the SLASL at E2 was more related to SLW at E1 than to SLW at E2 and the response in the Bohai Bay was not attributed to the local meteorological forcing. The AP and 10 m wind in the ERA-Interim data were spatially averaged in the Bohai Bay (R1 area in Fig. 1b) and the results were taken as the regional meteorological forcing. The ERA-LASLs, at both E1 and E2, were mainly forced by the ERA-LW at 75°; besides, the correlation coefficient between ERA-LASL at both E1 and E2 and ERA-LW at 75° was much larger than

that between local SLASL and local wind.

Based on the regional meteorological forcing, including ERA-LAP and ERA-LW at 75°, the IBR was implemented to estimate the response of the ocean to the forcing at E1 and E2. The MAE between the estimated and observed response was



13.13 cm (13.44 cm) and FO was 72.33% (71.78%) at E1 (E2), indicating that the estimated response was much closer to the observations than those obtained using the other methods, including IB correction, DAC, MLR with Eq. (5) and IBR with local forcing, as shown in Table 3, Table 5 and Fig. 7 (Fig. 8). The aforementioned results demonstrated that the developed new methodology IBR was a feasible and relatively effective method to estimate the response of the coastal ocean to the

5  meteorological forcing. Besides, most of the extreme events were influenced by the remote wind that was not considered in the IBR, so the estimated results using IBR with regional meteorological forcing deviated greatly from the observed values.

**Data availability**

The ERA-Interim data is downloaded from http://apps.ecmwf.int/datasets/. The HYCOM global analysis data is available at http://hycom.org. The in-situ observations used in this study will be made available on request.

10  **Appendix A. Brief glossary of the acronyms in this paper**

| | |
|---|---|
| ASL | adjusted sea level |
| AP | atmospheric pressure |
| DAC | dynamic atmospheric correction |
| ERA-ASL | adjusted sea level obtained using the regionally averaged ERA-Interim data |
| ERA-LAP | low-pass atmospheric pressure obtained from the regionally averaged ERA-Interim data |
| ERA-LASL | low-pass adjusted sea level obtained using the regionally averaged ERA-Interim data |
| ERA-LW | low-pass wind obtained from the regionally averaged ERA-Interim data |
| FO | the frequency of occurrences when the estimated values was within 0.15 m from the observed values |
| IB | inverted barometer |
| IBR | the method combined IB correction and regression analysis |
| LAP | low-pass atmospheric pressure |
| LASL | low-pass adjusted sea level |
| LSL | low-pass sea level |
| LUW | low-pass $u$ wind |
| LVW | low-pass $v$ wind |
| LW | low-pass wind |
| MAE | mean absolute error |
| MLR | multivariable linear regression |
| RSV | residual signal variance |
| RVR | ratio of the variation reduction |



| SAP | sub-sampled atmospheric pressure |
|---|---|
| SL | sea level |
| SLAP | sub-sampled low-pass atmospheric pressure |
| SLASL | sub-sampled low-pass adjusted sea level |
| SLSL | sub-sampled low-pass sea level |
| SLUW | sub-sampled low-pass $u$ wind |
| SLVW | sub-sampled low-pass $v$ wind |
| SLW | sub-sampled low-pass wind |
| SW | sub-sampled wind |

## Acknowledgements

The authors would like to thank Carlos Adrian Vargas Aguilera (nubeobscura@hotmail.com) for sharing the matlab code of cosine-lanczos filter and Zheng Guo for polishing the manuscript.

This work was supported by the National Key Research and Development Program of China (Grant No. 2017YFC1404700), the Discipline Layout Project for Basic Research of Shenzhen Science and Technology Innovation Committee (Grant No. 20170418), the Guangdong Special Fund Program for Economic Development (Marine Economic) (Grant No. GDME-2018E001), and Open Fund of Key Laboratory of Engineering Sediment of the Ministry of Transport, Tianjin Research Institute for Water Transport Engineering, M.O.T.

## Refersences

Amin, M.: On analysis and forecasting of surges on the west coast of Great Britain, On analysis and forecasting of surges on the west coast of Great Britain, 68, 79-94, 1982.

Andres, M., Gawarkiewicz, G. G., and Toole, J. M.: Interannual sea level variability in the western North Atlantic: Regional forcing and remote response, Geophysical Research Letters, 40, 5915-5919, 2013.

Calafat, F. and Chambers, D.: Quantifying recent acceleration in sea level unrelated to internal climate variability, Geophysical Research Letters, 40, 3661-3666, 2013.

Carrère, L. and Lyard, F.: Modeling the barotropic response of the global ocean to atmospheric wind and pressure forcing-comparisons with observations, Geophysical Research Letters, 30, 2003.

Carrere, L., Faugère, Y., and Ablain, M.: Major improvement of altimetry sea level estimations using pressure-derived corrections based on ERA-Interim atmospheric reanalysis, Ocean Science, 12, 825-842, 2016.

Castro, B. M. and Lee, T. N.: Wind-forced sea level variability on the southeast Brazilian shelf, Journal of Geophysical Research: Oceans, 100, 16045-16056, 1995.

Dee, D. P., Uppala, S., Simmons, A., Berrisford, P., Poli, P., Kobayashi, S., Andrae, U., Balmaseda, M., Balsamo, G., Bauer, d. P., and Beljaars, A.: The ERA-Interim reanalysis: Configuration and performance of the data assimilation system, Quarterly Journal of the Royal Meteorological Society, 137, 553-597, 2011.

Duchon, C. E.: Lanczos filtering in one and two dimensions, Journal of Applied Meteorology, 18, 1016-1022, 1979.

Egbert, G. D. and Erofeeva, S. Y.: Efficient Inverse Modeling of Barotropic Ocean Tides, Journal of Atmospheric & Oceanic Technology, 19, 183-204, 2002.

Fukumori, I., Raghunath, R., and Fu, L. L.: Nature of global large-scale sea level variability in relation to atmospheric forcing: A modeling study, Journal of Geophysical Research: Oceans, 103, 5493-5512, 1998.



Grinsted, A., Moore, J. C., and Jevrejeva, S.: Application of the cross wavelet transform and wavelet coherence to geophysical time series, Nonlinear Processes in Geophysics, 11, 561-566, 2004.

Hirose, N., Fukumori, I., Zlotnicki, V., and Ponte, R. M.: Modeling the high-frequency barotropic response of the ocean to atmospheric disturbances: Sensitivity to forcing, topography, and friction, Journal of Geophysical Research: Oceans, 106, 30987-30995, 2001.

Hsueh, Y. and Romea, R. D.: Wintertime winds and coastal sea-level fluctuations in the northeast China Sea. Part I: Observations, Journal of physical oceanography, 13, 2091-2106, 1983.

Kurapov, A. L., Erofeeva, S. Y., and Myers, E.: Coastal sea level variability in the US West Coast Ocean Forecast System (WCOFS), Ocean Dynamics, 67, 23-36, 2017.

Li, K. and Yang, K.: The non-periodic sea level vatiation in relation to wind and pressure at the Bohai Bay, Marine Sciences, 2, 12-15,
10    1983.

Lv, X., Wang, D., Yan, B., and Yang, H.: Coastal sea level variability in the Bohai Bay: influence of atmospheric forcing and prediction, Journal of Oceanology and Limnology, doi: 10.1007/s00343-019-7383-y, 2018. Accepted, 2018.

Marshall, J., Hill, C., Perelman, L., and Adcroft, A.: Hydrostatic, quasi-hydrostatic, and nonhydrostatic ocean modeling. J Geophys Res 102(C3):5733-5752, Journal of Geophysical Research Atmospheres, 102, 5733-5752, 1997.

Melet, A., Almar, R., and Meyssignac, B.: What dominates sea level at the coast: a case study for the Gulf of Guinea, Ocean Dynamics, 66, 623-636, 2016.

Melet, A., Meyssignac, B., Almar, R., and Le Cozannet, G.: Under-estimated wave contribution to coastal sea-level rise, Nature Climate Change, 8, 234, 2018.

Militello, A. and Kraus, N. C.: Generation of harmonics by sea breeze in nontidal water bodies, Journal of physical oceanography, 31,
20    1639-1647, 2001.

MWRPRC: Bulletin of Chinese rivers and sediments 2014, China Water Power Press, Beijing, 2014.

Paraso, M. C. and Valle-Levinson, A.: Meteorological Influences on Sea Level and Water Temperature in the Lower Chesapeake Bay: 1992, Estuaries, 19, 548-561, 1996.

Piecuch, C. G., Dangendorf, S., Ponte, R. M., and Marcos, M.: Annual sea level changes on the North American Northeast Coast:
influence of local winds and barotropic motions, Journal of Climate, 29, 4801-4816, 2016.

Sandstrom, H.: On the wind-induced sea level changes on the Scotian shelf, Journal of Geophysical Research: Oceans, 85, 461-468, 1980.

Spitz, Y. and Klinck, J. M.: Estimate of bottom and surface stress during a spring-neap tide cycle by dynamical assimilation of tide gauge observations in the Chesapeake Bay, Journal of Geophysical Research Oceans, 103, 12761-12782, 1998.

Thompson, P. R., Merrifield, M. A., Wells, J. R., and Chang, C. M.: Wind-driven coastal sea level variability in the northeast pacific,
Journal of Climate, 27, 4733-4751, 2014.

Tierney, C., Wahr, J., Bryan, F., and Zlotnicki, V.: Short-period oceanic circulation: Implications for satellite altimetry, Geophysical Research Letters, 27, 1255-1258, 2000.

Tilburg, C. E. and Garvine, R. W.: A Simple Model for Coastal Sea Level Prediction, Weather and Forecasting, 19, 511-519, 2004.

Vinogradova, N. T., Ponte, R. M., and Stammer, D.: Relation between sea level and bottom pressure and the vertical dependence of
oceanic variability, Geophysical Research Letters, 34, 2007.

Wang, Y., Li, Y., Liu, W., and Gao, Y.: Assessing operational ocean observing equipment (OOOE) based on the fuzzy comprehensive evaluation method, Ocean Engineering, 107, 54-59, 2015.

Wunsch, C. and Stammer, D.: Atmospheric loading and the oceanic "inverted barometer" effect, Reviews of Geophysics, 35, 79-107, 1997.

Zhang, N., Wang, J., Wu, Y., Wang, K.-H., Zhang, Q., Wu, S., You, Z.-J., and Ma, Y.: A modelling study of ice effect on tidal damping in
the Bohai Sea, Ocean Engineering, 173, 748-760, 2019.

Zhao, B. and Cao, D.: Wintertime low frequency fluctuations of Chinese coastal sea-level in the Huanghai Sea and the East China Sea, OCEANOLOGIA ET LIMNOLOGIA SINICA, 18, 563-574, 1987.



**Table 1. The mean absolute error (MAE) and the correlation coefficients (R) between the meteorological forcing in the sub-sampled in-situ observations and the ERA-Interim data**

| Meteorological forcing | E1 | | E2 | |
|---|---|---|---|---|
| | MAE | R | MAE | R |
| AP | 0.85 mbar | 1.00 | 0.50 mbar | 1.00 |
| $u$ wind | 1.63 m/s | 0.82 | 2.28 m/s | 0.75 |
| $v$ wind | 1.61 m/s | 0.82 | 2.48 m/s | 0.81 |

**Table 2. Regression coefficients as shown in Eq. (2) at E1 and E2, when the local and regional meteorological forcing were used**

| Station | Method | $\alpha_0$ (m) | $\alpha_1$ (m/mbar) | $\alpha_2$ (s) |
|---|---|---|---|---|
| E1 | IBR_local | 6.87 | $-6.71\times10^{-3}$ | $-8.34\times10^{-2}$ |
| | IBR_regional | 9.63 | $-9.36\times10^{-3}$ | $-4.90\times10^{-2}$ |
| E2 | IBR_local | 6.44 | $-6.26\times10^{-3}$ | $-1.11\times10^{-2}$ |
| | IBR_regional | 8.06 | $-7.83\times10^{-3}$ | $-4.47\times10^{-2}$ |



**Table 3. Dominant wind orientation, mean absolute errors (MAEs) and the correlation coefficient (R) between the estimated and observed SLSL, and FO when different methods were used at E1 and E2**

| Location | Method | Dominant wind orientation (°) | Estimation of SLSL | | |
|---|---|---|---|---|---|
| | | | R | MAEs (cm) | FO[1] (%) |
| E1 | IB | --- | 0.35 | 20.13 | 48.22 |
| | DAC[2] | --- | 0.70 | 18.18 | 51.51 |
| | MLR with Eq. (5) | 65 | 0.62 | 14.51 | 66.03 |
| | IBR_local | 80 | 0.65 | 13.76 | 67.40 |
| | IBR_regional | 75 | 0.70 | 13.13 | 72.33 |
| E2 | IB | --- | 0.31 | 19.28 | 47.40 |
| | DAC[2] | --- | 0.63 | 17.42 | 54.25 |
| | MLR with Eq. (5) | 55 | 0.50 | 16.03 | 58.08 |
| | IBR_local | 10 | 0.36 | 17.10 | 57.53 |
| | IBR_regional | 75 | 0.65 | 13.44 | 71.78 |

[1] The frequency of occurrences when the estimated SLSL within 0.15 m from the observed SLSL.

5  [2] The best performance of DAC was obtained in Exp7 on the whole, so the estimated results in Exp7 were taken as the results of DAC.





**Table 4. The detailed model settings of the numerical experiments, when the DAC was used**

| No. | Meteorological forcing | Dimension | Horizontal resolution (′) | Temperature and salinity |
|---|---|---|---|---|
| Exp1-tide | No | 2D | 7.5 | TS1 |
| Exp2-tide | No | 2D | 2 | TS1 |
| Exp1 | AP | 2D | 7.5 | TS1 |
| Exp2 | AP+Wind | 2D | 7.5 | TS1 |
| Exp3 | AP | 2D | 2 | TS1 |
| Exp4 | AP+Wind | 2D | 2 | TS1 |
| Exp5 | AP | 3D | 7.5 | TS1 |
| Exp6 | AP | 3D | 7.5 | TS2 |
| Exp7 | AP+Wind | 3D | 7.5 | TS1 |
| Exp8 | AP+Wind | 3D | 7.5 | TS2 |
| Exp9 | AP | 3D | 2 | TS1 |
| Exp10 | AP+Wind | 3D | 2 | TS2 |



**Table 5. Residual signal variance [Var(Obs-estimated), cm$^2$] and ratio of the variation reduction compare to IB [Var(Obs-IB)-Var(Obs-estimated)]/Var(Obs-IB), when different methods were used**

|  | RSV_E1[1] (cm$^2$) | RSV_E2[2] (cm$^2$) | RVR_E1[3] (%) | RVR_E2[4] (%) |
|---|---|---|---|---|
| Obs | 688.39 | 643.17 | --- | --- |
| Obs−IB | 612.73 | 582.49 | 0 | 0 |
| Obs−Exp1 | 607.16 | 574.39 | 0.91 | 1.39 |
| Obs−Exp2 | 471.04 | 467.23 | 23.12 | 19.79 |
| Obs−Exp3 | 605.69 | 572.63 | 1.15 | 1.69 |
| Obs-Exp4 | 471.01 | 463.99 | 23.13 | 20.34 |
| Obs−Exp5 | 605.82 | 573.42 | 1.13 | 1.56 |
| Obs−Exp6 | 606.08 | 573.64 | 1.09 | 1.52 |
| Obs−Exp7 | 462.21 | 464.27 | 24.57 | 20.30 |
| Obs−Exp8 | 462.44 | 464.59 | 24.53 | 20.24 |
| Obs−Exp9 | 604.73 | 571.80 | 1.31 | 1.84 |
| Obs−Exp10 | 467.77 | 461.88 | 23.66 | 20.71 |
| Obs−MLR with Eq. (5) | 425.79 | 479.42 | 30.51 | 17.69 |
| OBS−IBR_local | 401.45 | 561.70 | 34.48 | 3.57 |
| Obs−IBR_regional | 353.71 | 368.20 | 42.27 | 36.79 |

[1] Residual signal variance at E1

[2] Residual signal variance at E2

[3] Ratio of the variation reduction compare to IB at E1

[4] Ratio of the variation reduction compare to IB at E2





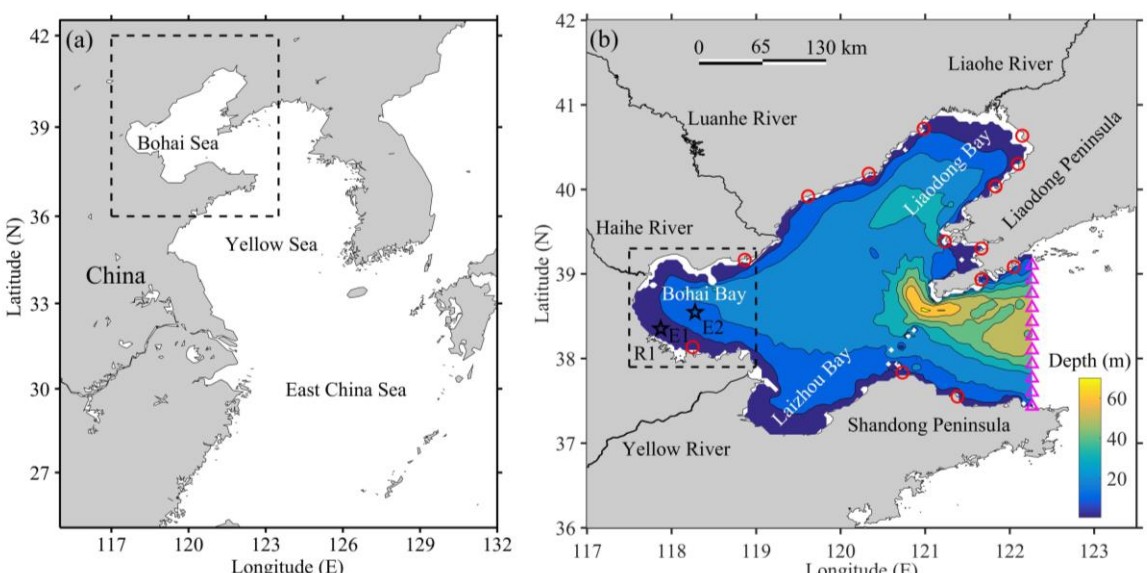

**Figure 1. (a) Map showing the general location of the Bohai Sea (rectangle with dotted lines). (b) Map showing the locations of observation stations (black stars), E1 and E2, in the Bohai Bay; the location of the tide gauge stations (red circles) and the east open boundary (magenta triangles); the area R1 (rectangle with dotted lines), where the ERA-wind is spatially averaged; and bathymetry of the Bohai Sea (colors).**

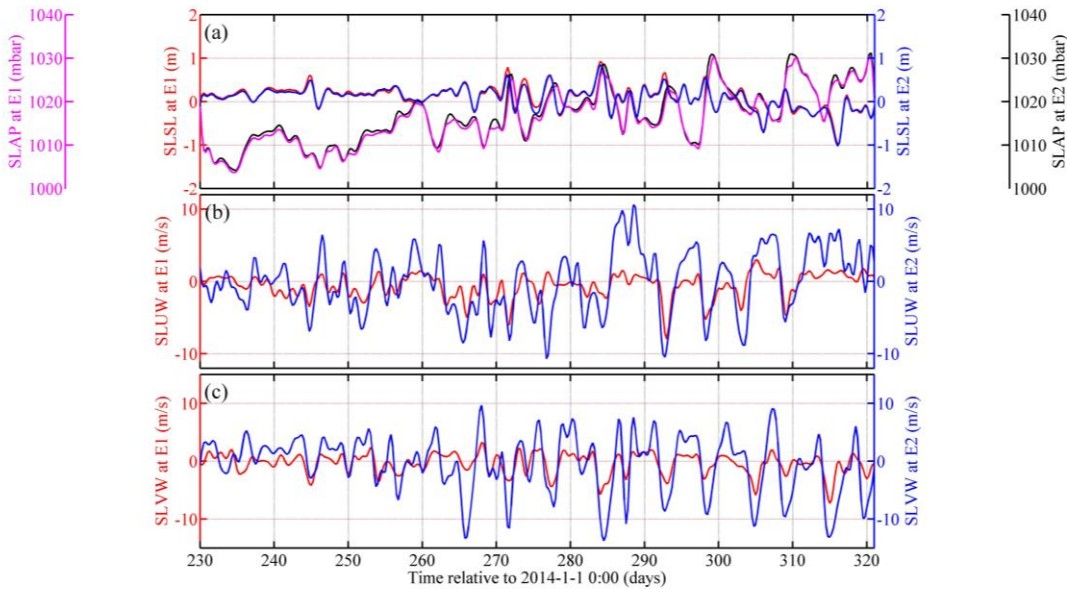

**Figure 2. (a) Time series of SLAP at E1 (magenta line), SLSL at E1 (red line), SLAP at E2 (black line) and SLSL at E2 (blue line). (b) Time series of SLUW at E1 (red line) and E2 (blue line). (c) Same as (b), but for SLVW.**




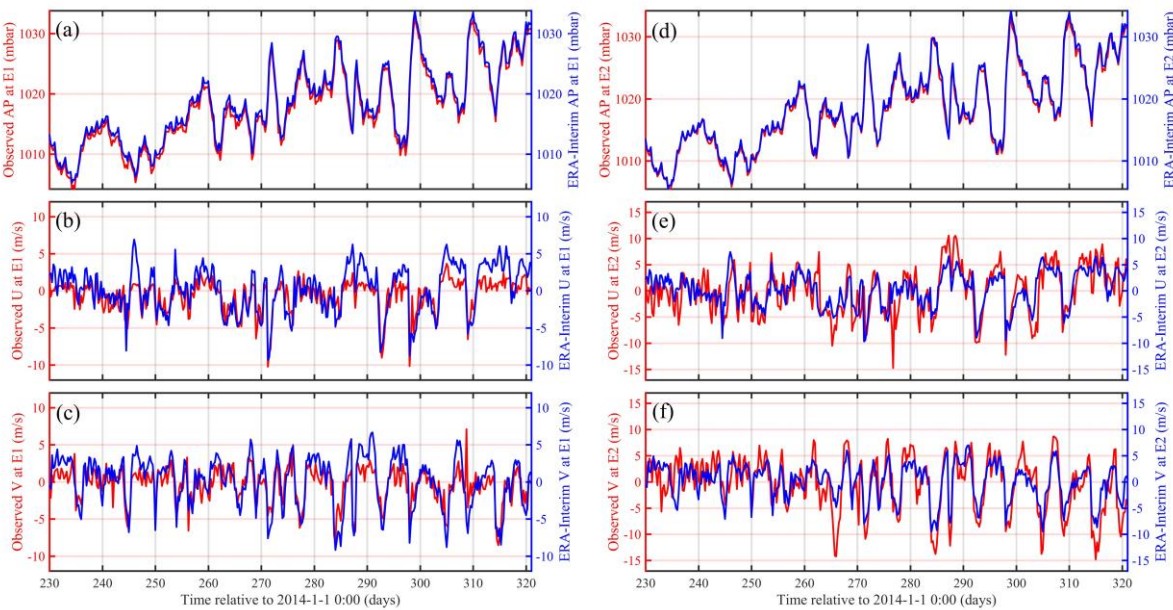

**Figure 3. (a) Time series of the sub-sampled in-situ observations of AP at E1 (red line) and the six-hourly AP at E1 in ERA-Interim data (blue line); (b) same as (a) but for *u* wind component; (c) same as (a) but for *v* wind component; (d-f) same as (a-c) but for those at E2.**

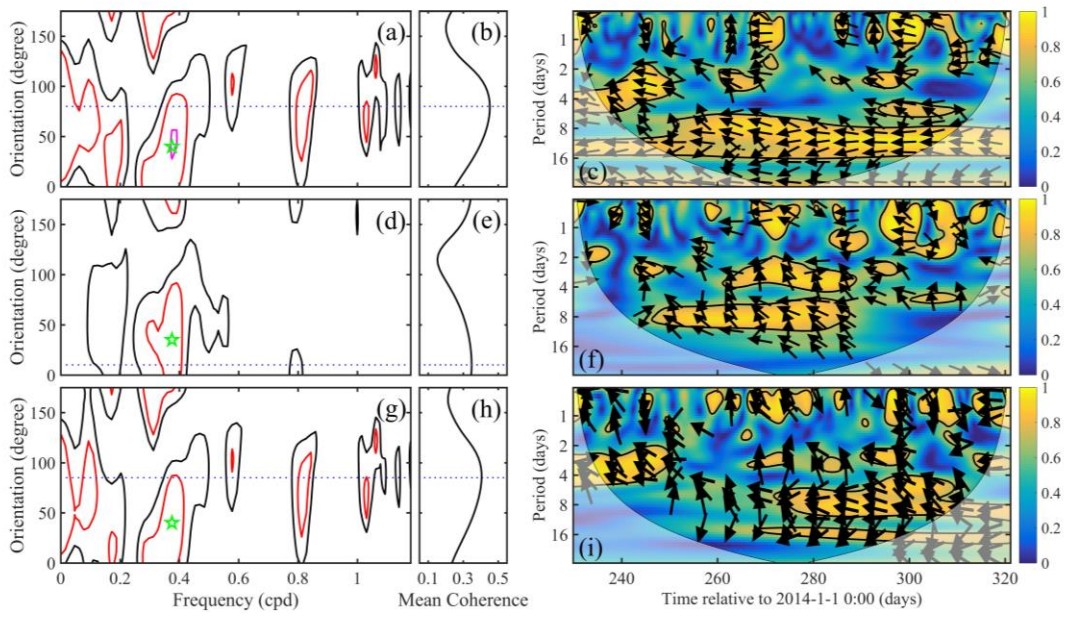

**Figure 4. (a) Contours of magnitude squared coherence between SLASL and SLW at E1 as a function of frequency and wind orientation (in degree measured clockwise from north at an interval of 5 degrees), where the values are 0.5**





(black line), 0.7 (red line) and 0.9 (magenta line), and the location of the maximum value (green star). (b) Contours of the averaged magnitude squared coherence between SLASL and SLW at E1 at every wind orientation. Blue dotted lines in (a) and (b) show the wind orientation, where the averaged magnitude squared coherence reaches the maximum. (c) The wavelet coherence (colors) and phase (arrows) between SLASL at E1 and SLW at E1, when the averaged magnitude squared coherence reaches the maximum, in which the 5% significance level against red noise (thick black contour) and the cone of influence where edge might distort the picture (lighter shade) are indicated. (d-f) Same as (a-c), but for those between SLASL and SLW at E2; (g-i) Same as (a-c), but for those between SLASL at E2 and SLW at E1.

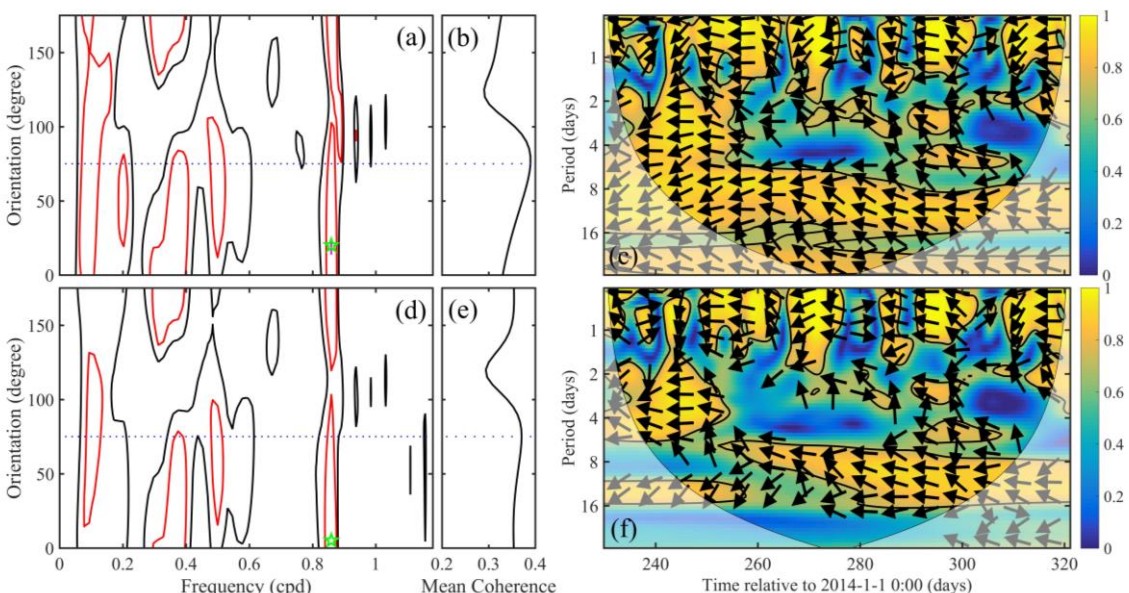

Figure 5. Same as Figure 4, but for those between ERA-LASL and ERA-LW at (a-c) E1 and (d-f) E2.




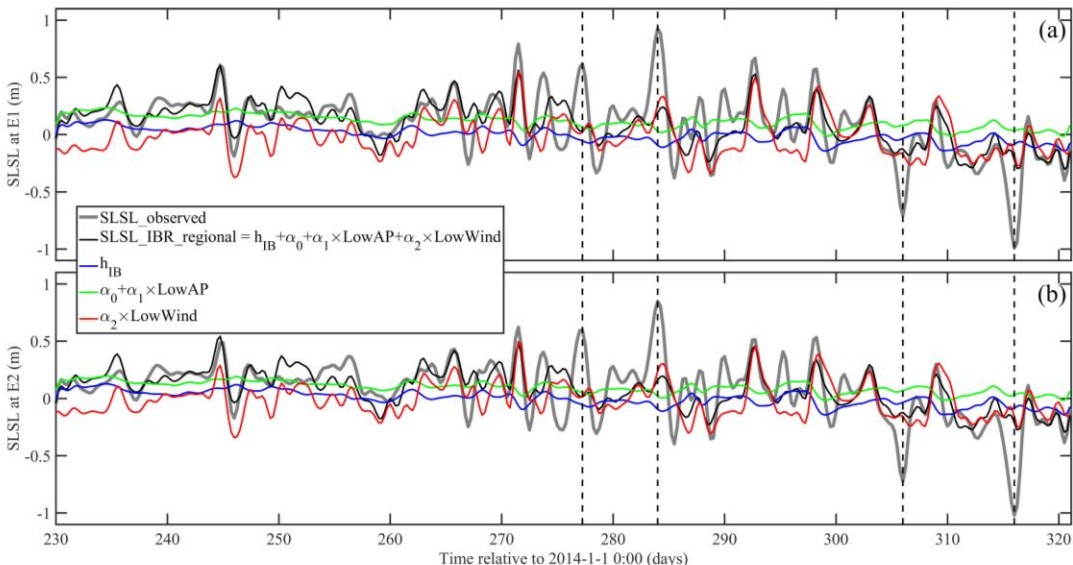

**Figure 6. The observed values of SLSL (gray line), the estimated SLSL using IBR with the regional meteorological forcing (black line) and the components, at (a) E1 and (b) E2. The black dotted lines show the extreme events.**

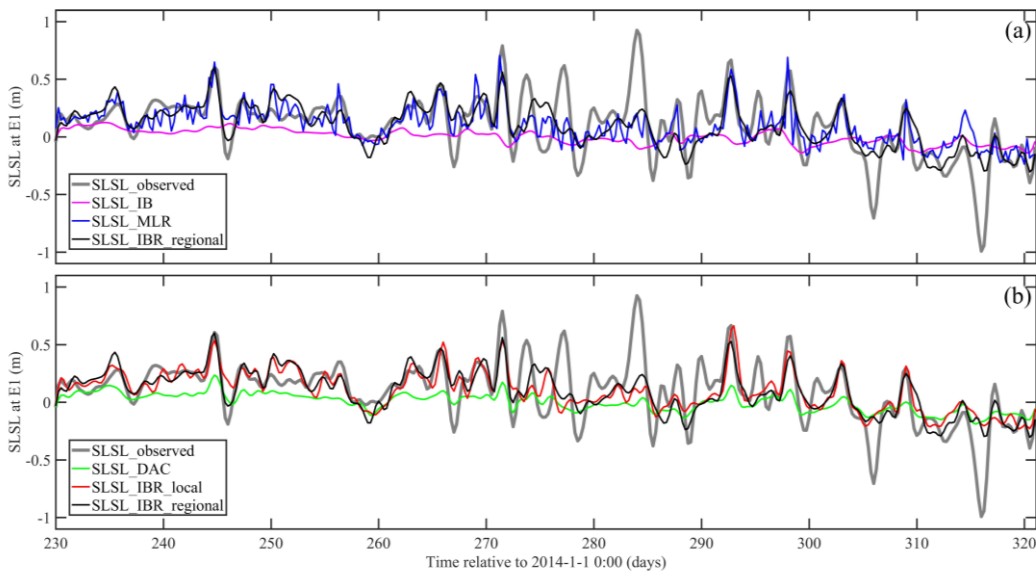

**Figure 7. (a) Time series of the observed SLSL (gray line) and the corresponding estimated values using IB (magenta line), MLR with Eq. (5) (blue line) and IBR with regional meteorological forcing (black line); (b) time series of the observed SLSL (gray line) and the corresponding estimated values using DAC (green line), IBR with local meteorological forcing (red line) and IBR with regional meteorological forcing (black line), at E1.**





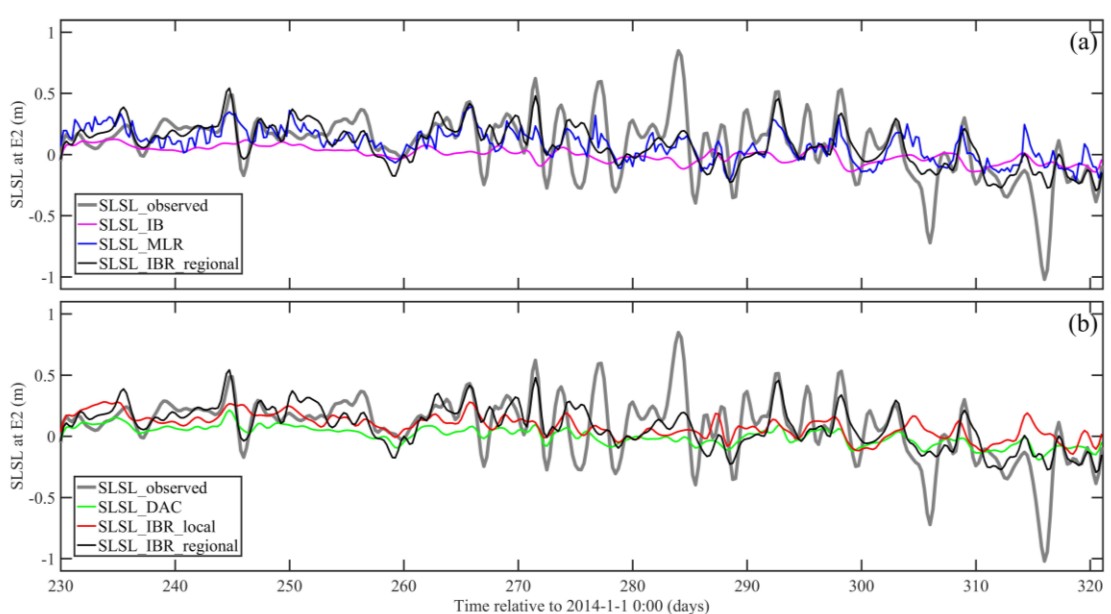

**Figure 8. Same as Figure 7, but for those at E2.**

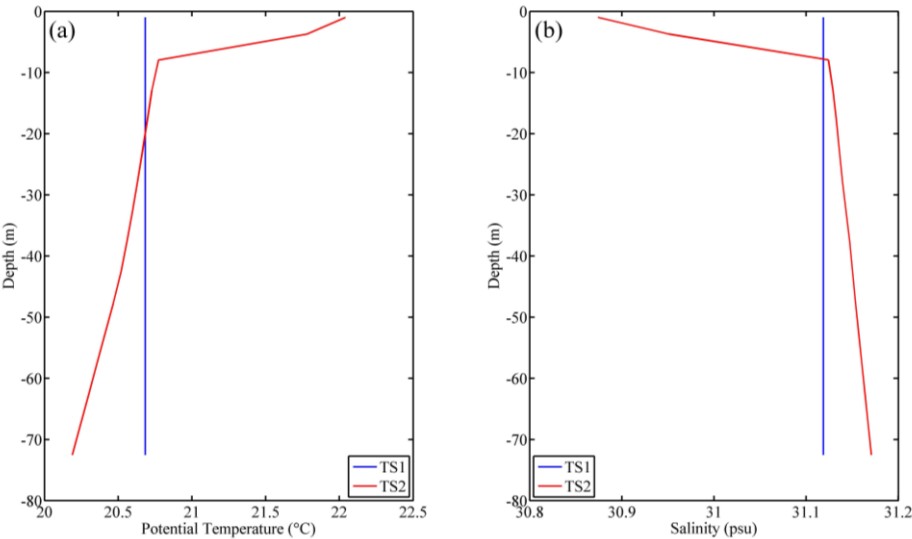

**Figure 9. Horizontally homogeneous profiles of initial (a) potential temperature and (b) salinity used in the numerical experiments, including TS1 (blue lines) and TS2 (red lines), which were extracted from the HYCOM global analysis results.**




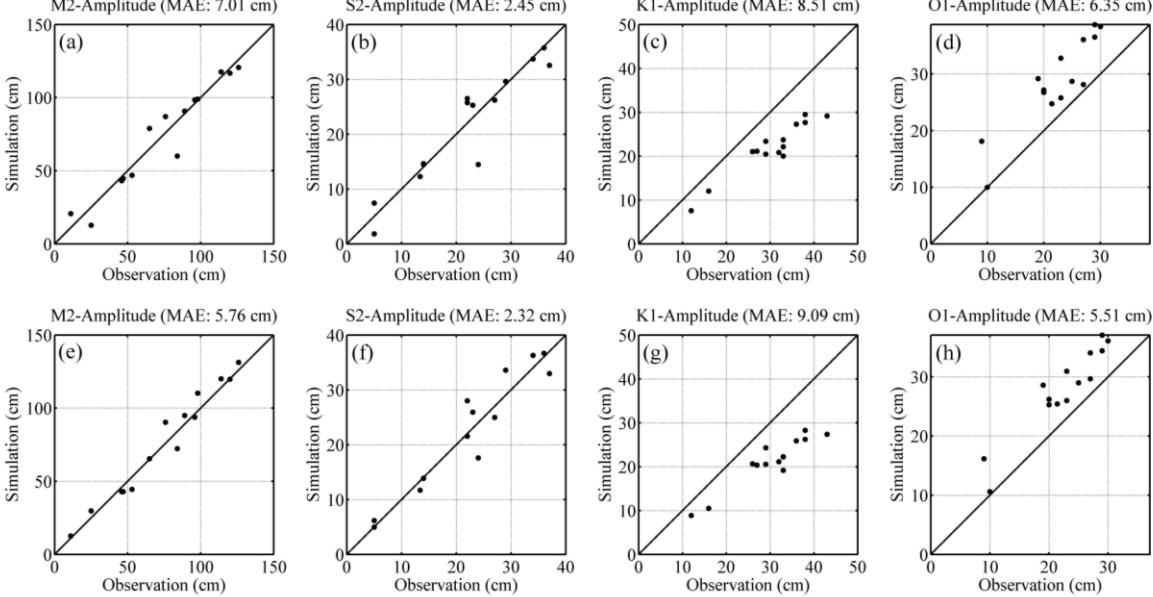

**Figure 10. Comparison of simulated and observed amplitude for (a) $M_2$, (b) $S_2$, (c) $K_1$ and (d) $O_1$ in numerical experiment Exp1-tide, in which the horizontal resolution is 7.5′. (e-h) Same as (a-d), but for those in numerical experiment Exp2-tide, in which the horizontal resolution is 2′.**

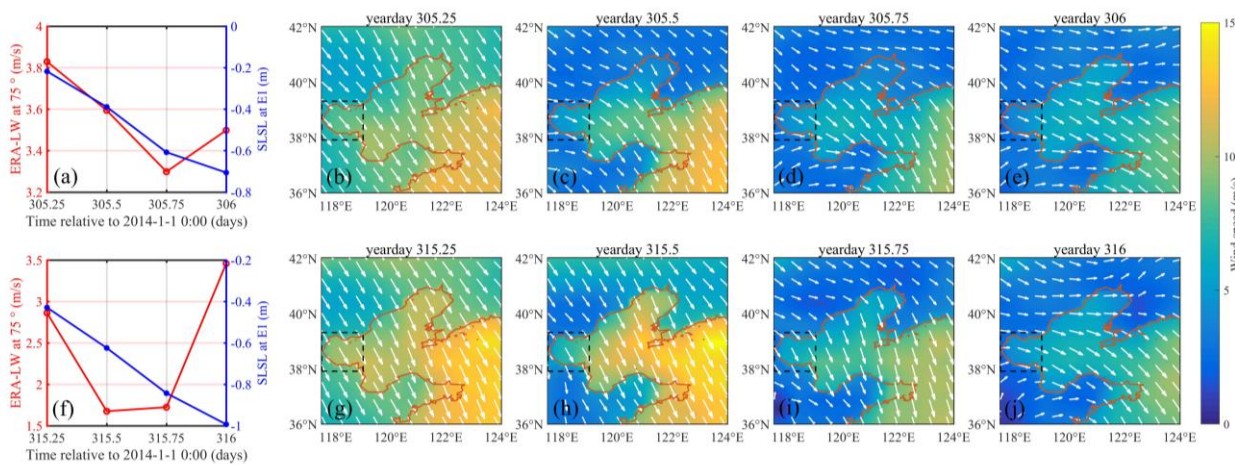

**Figure 11. (a) Time series of ERA-LW at 75° (red line) and SLSL at E1 (blue line) from yearday 305.25 to 306; (b) the spatial distributions of the wind speed (colors) and direction (white arrows) at yearday 305.25; the Bohai Bay was shown by the rectangle with dotted lines; and (c-e) same as (b), but for those at yeardays 305.5, 305.75 and 306, respectively. (f-j) Same as (a-e), but for those from yearday 315.25 to 316.**





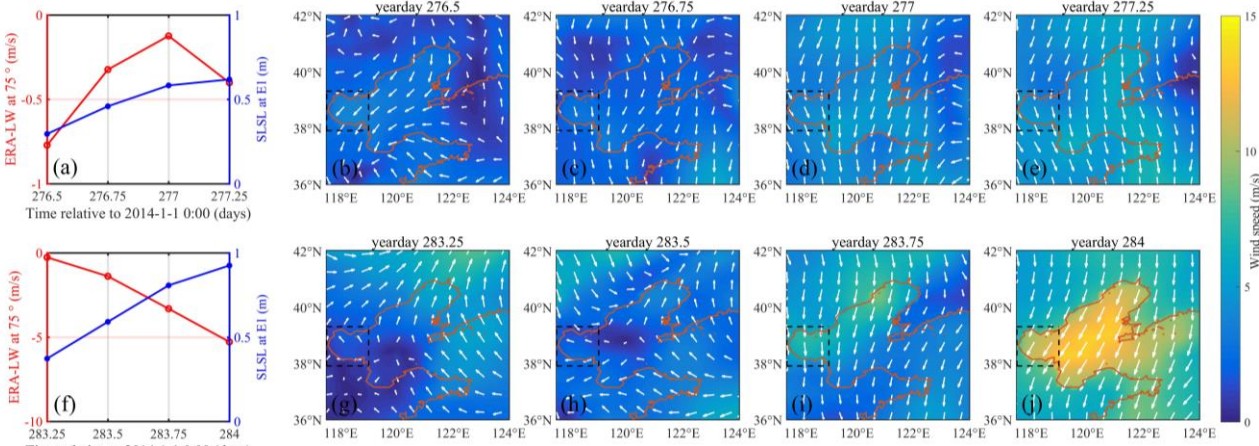

**Figure 12. Same as Figure 11, but for those (a-e) from yearday 276.5 to 277.25 and (f-j) from yearday 283.25 to 284.**

