# Peer review of "A methodology for estimating the response of the coastal ocean to meteorological forcing: A case study in the Bohai Bay"

_Ocean Science, 2019_

## Referee Comment (RC1) · Anonymous Referee #1 · 13 Jun 2019

This study evaluates the relative success of different parameterisations of the static and dynamic response to meteorological forcing of sea level at two locations in Bohai Bay. The study tests the classic Inverse Barometer Correction, dynamic atmospheric correction (DAC), multivariable linear regression analysis and a new approach (the IBR) that combines the high frequency dynamic adjustment to atmospheric pressure with low frequency atmospheric pressure and wind components. This IBR adjustment is attempted using alternative regional and local atmospheric forcings and finds the regional IBR to afford the closest resemblance to sea level observations at the two locations.

[Figure]

General comments: The paper is well written and the authors describe a clear and methodical approach. In that respect, I find there is little to fault in the quality of their analysis, but the study does lack some context, as there is little description of the local environment, climate and sea level variability. For example, does this area suffer from frequent storm surges and inundation? Are tidal ranges large, so that the combined effects of meteorological and astronomical forcings have been particularly damaging here? Without this context, it leaves the reader rather underwhelmed and wondering what is the relevance/importance of this study to the local area and to the wider scientific community?

A few further comments: P1 Line 10 (Abstract) Perhaps replace "substantially contributed" with "dominated by"?

P 1 line 29 the second use of "sea level" can be abbreviated to "SL"

P2 line 4 replace "which makes the response be poorly accounted for" with "which is a poor representation of the response"

P4 line 14. The reported accuracy of the instrument (+/- 5cm) concerns me as this study is evaluating cm-scale sea level responses. GLOSS standards recommend the use of data with instruments with accuracy of +/- 1cm. Given this reported accuracy, it would be helpful to know whether any quality assurance processing had taken place (and if so what) using the underlying sea level observations.

P5 line 9 (and elsewhere in the paper) Confidence levels are not given for the correlations. These should be specified in the text and tables.

P6 line 17 "regional ASL" is misleading and would be better described as "regionally adjusted SL"

P7 line 10 replace "smaller" with "lower"

P8 line 5 should read "As the ocean has a dynamic response"

---

## Short Comment (SC1) · 30 Sep 2019

The paper presents a multivariable linear regression method (IBR) to estimate the effect of atmospheric conditions on the sea level variations. The skill of the method is demonstrated for two selected stations in the Bohai Sea in northeastern China. Unfortunately, I cannot recommend publishing this manuscript in Ocean Science, because both the method and also the obtained results are not acceptable.

The method is far from being new or innovative. It is a standard multivariable regression method, just using regional low-pass filtered input data. However, it remains totally unclear, how these results depend on the actual size of the selected area to determine

the regional values. If one increases the size of the regional window, more far-field effects would be accounted for. Probably there is a strong dependency of the method on the specific conditions in the area of interest, but perhaps also on other factors, e.g., the season of the year. This brings me to the other major criticism regarding the method. As it is designed now, it is very site-specific. It is not even clear, whether for other regions the standard multilinear regression method could provide better results. Any discussion regarding this issue is missing.

In addition, the obtained results do not justify a publication. The only "real" results related to the underlying physical processes in the Bohai Sea is the conclusion that the regional forcing has a stronger impact on the sea level than the local forcing. As already stated above, this strongly reflects the specific conditions in the Bohai Sea. Furthermore, even for two different stations in the Bohai Sea, the dominance of the regional forcing is quite different as shown in the paper. A simple two-dimensional numerical storm surge model could provide the dynamically correct response of the atmospheric forcing on the sea level variations for the entire Bohai Sea, which would allow a much more sound analysis of the underlying physical processes. For the 2D case, the mentioned limitations due to the too coarse horizontal resolution of the topography should be no problem. When forced with adequate open boundary conditions from a larger open ocean storm surge model, also external surges could be resolved, which is a major culprit of the IBR method presented here. However, as can be seen from figs. 7 and 8, such realistic open boundary conditions were not employed for the model study carried out in this paper. Therefore, the comparison of the model skill with the skill of the IBR method is extremely questionable.

I have several other criticisms regarding minor issues. However, at the present stage, it would make no sense to list them all. Last but least, I want to mention that altogether six authors from four different institutions sign responsible for this paper. Considering this, a more sound scientific work would be expected.

---

## Referee Comment (RC2) · Anonymous Referee #2 · 7 Oct 2019

Review: A methodology for estimating the response of the coastal ocean to meteorological forcing: a case study in the Bohai Bay, by D. Wang et al.

This study aims at developing a methodology to estimate the local sea level (SL) response to meteorological forcing, including the static and dynamic ocean response to varying surface atmospheric pressure (AP) and winds. The method, called IBR, is based on the inverted barometer effect and a multilinear regression (MLR) of total sea level to local or regional surface atmospheric pressure and winds at the dominant direction.

Although I appreciate that substantial work has been performed, I recommend the

paper not to be published in Ocean Science. I provide below the main underlying reasons, and will not discuss specific, more minor issues with the paper.

First, the two selected locations for which the IBR method has been developed and tested are quite peculiar, located inside the shallow, semi-enclosed Bohai Sea. Due to its shallow depth and location at mid-latitudes, the ocean dynamics should be prone to small space and time scales ocean dynamicsÂăand thus to larger deviations of the SL to meteorological forcing from the static, inverted barometer (IB) response only (e.g. Carrère and Lyard 2003). In this region, the annual variance of the IB effect is also quite pronounced (e.g. Ponte 2006) compared to other regions worldwide. Thus, the conclusions of the study are not expected to hold for other locations. A short 3-month period is analysed here. Already over this period the IBR fails at reproducing different events. The reader can wonder whether the MLR would work over longer periods, especially when the dominant wind direction changes over time, or when remote forcing, not accounted for here, might be more dominant. The main result could be that regional wind and atmospheric surface pressure forcing are more important than the local forcings in determining the local SL response to meteorological forcing. Yet, no justification is given for the selection of the regional area used to compute regional winds and atmospheric surface pressure, nor a sensitivity to the regional area provided.

The purpose of the method is not completely clear to me. Local sea level variations result from different processes, amongst which are the static dynamical response of the SL to meteorological forcing. Other processes contribute, and it is not clear how using an MLR with the total SL as the response variable and only surface wind and surface atmospheric pressure as explanatory variables would be adequate (e.g. overfitting issues, interpretation of regression coefficients in a context of co-variations between driving processes, including other processes than considered here, etc). On longer periods, notably, other explanatory variables should be used, to account for changes in SL due to e.g. the general ocean circulation. Estimating only the isostatic response of the ocean to atmospheric pressure forcing is useful if one is only interested in a

dynamical interpretation of sea level records.

In addition, the IBR does not really represent a real innovative approach. Thompson (1986) and Woodworth (1987) already used a multiple linear regression on both AP and surface winds. The added value of IBR compared to classical dynamical atmospheric corrections infered from hydrodynamic models such as MOG2D-G, which provides a process-based, physical correction for targeted processes, is not compelling. In hydrodynamic models, not only local or regional effects are accounted for, but also the high-frequency barotropic SL response to remote fast and large scale atmospheric forcing. I appreciate the simulations performed with HYCOM to compare the results of the IBR with a modelled DAC using a hydrodynamic model in a barotropic or baroclinic configuration. However, the model settings, and experiments are not presented in a convincing way. It would have been interesting to compare the MLR results to the DAC used in different altimetric products, such as that provided by MOG2D-G for the altimetric data produced by SSALTO-DUACS.

Ponte RM (2006) Low-Frequency Sea Level Variability and the Inverted Barometer Effect. Journal of Atmospheric and Oceanic Technology, 23, 619-629.